# (18-Crown-6)-bis(tetrahydrofuran)-potassium Anthracenide: The Salt of a Free Radical Anion Crystallizing as a Kryptoracemate

**Ivan Bernal [1,2] and Roger A. Lalancette [1,*]**

1   Carl A. Olson Memorial Laboratories, Department of Chemistry, Rutgers University, 73 Warren St., Newark, NJ 07102, USA; bernalibg@gmail.com
2   Molecular Sciences Institute, School of Chemistry, University of the Witwatersrand, Private Bag 3, Johannesburg 2050, South Africa
*   Correspondence: roger.lalancette@gmail.com; Tel.: +1-973-353-5646

**Abstract:** K[(18-crown-6)-bis(tetrahydrofuran)anthracenide] was independently prepared by three groups, and its structure described by two of them. The third structure description, though listed in the Cambridge Crystallographic Data Centre (CSD) collection, contains no space group or atomic coordinates, but the cell constants leave no doubt that it is the same species as the two others, which were reported in 2006 and 2016. The compound crystallizes in space group $P2_1$, with $Z' = 2.0$ at T = 123 K and R = 4.91% (**I**), and at 100 K and R = 4.44% (**II**); both impressive results in their individual quality and agreement, despite differences in experimental methods and the temperature of data collection. A more detailed examination of the published data for (**I**) and (**II**) reveals that the correct description for this very unusual, and thus far unique substance, is that it contains a radical anion crystallizing as a *kryptoracemate* rather than as a simple example of a Sohncke space group with $Z' = 2.0$. The anthracenide anions present in (**I**) and (**II**) are virtually identical; in contrast, the internal pair of cationic species differ from one another in the dissymmetry of the flexible tetrahydrofuran ligands, having significantly different internal and external torsional angles. The two THF molecules attached to the K(crown-ether) cations are not centrosymmetrically related, and this is what makes this portion of the asymmetric unit responsible for the crystal being a kryptoracemate. Our presentation will be based on the more fully documented sample (**II**), unless specifically stated.

**Keywords:** free radicals; radical anions; crown ethers; potassium cations; crown ether cations; sohncke space groups; enantiomorphs; kryptoracemic crystals; kryptoracemic pairs; molecular dissymmetry; crystallographic center of mass; asymmetric unit cell (Z') of crystals; molecular overlay diagrams

## 1. Introduction

The CSD [1] contains three entries labeled YETPAP [2], YETPAP01 [3], and YETPAP02 [4] describing the crystal structure of K[(18-crown-6)-bis(tetrahydrofuran)anthracenide]. The entry YETPAP01 [3] will be hereafter ignored, since it is described as disordered [1,2]. In YETPAP, Rosokha and Kochi [2] described their unusual findings thus: "Arene cation-radicals and anion-radicals result directly from the one-electron oxidation and reduction of many aromatic hydrocarbons, yet virtually nothing is known of their intrinsic (thermodynamic) stability and hence 'aromatic character'". Furthermore, they state: "Since such paramagnetic ion radicals lie intermediate between aromatic (Hückel) hydrocarbons with $4n + 2$ electrons and antiaromatic analogues with $4n$-electrons, we can now address the question of $\pi$-delocalization in these odd-electron counterparts. Application of the structure-based "harmonic oscillator model of aromaticity" or the HOMA method leads to the surprising conclusion that the aromaticity of these rather reactive, kinetically unstable arene cation and anion radicals (as measured by the HOMA index) is actually higher than

that of their (diamagnetic) parent contrary to conventional expectations". Finally, they go on to say that they encountered problems in the preparation and isolation of other alkali metal derivatives of this fascinating anthracenide cation. It was these remarks that led us to further search the published literature for additional information.

By contrast, sometime later, Castillo et al. [4] succeeded in isolating all the alkali metal derivatives and determined the structure of a number of them, of which the one that we selected is the one we label (**II**), which appears in the CSD [1] as YETPAP02 [4]. None of the other alkali metal derivatives belong in Sohncke space groups [4].

N.B. The programs used in this document are named and identified in references [5,6].

Definition: Kryptoracemic crystallization [7–11] is the phenomenon whereby a racemic solution produces crystals whose asymmetric unit contents are imperfect racemic pairs, and thus are relegated to a Sohncke space group. The most common reason for the existence of differing pairs is that dissymmetric features of flexible fragments on portions of the asymmetric unit (such as torsional angles) no longer obey symmetry operations of the second kind (e.g., inversions, mirror planes, etc.), as required in the case of a simple racemate [7–11].

Below is a simplified description of kryptoracemic crystallization.

Racemic ($\pm$)-[Co(en)$_3$]I$_3$. H$_2$O (en = ethylenediamine) was the first compound Alfred Werner separated into its antipodes [1], and whose optical activity was not a direct property of one or more chiral atomic centers but the result of the dissymmetry at the metal due to the helical arrangement of its three bidentate ligands, as in the case of the [Co(oxalato)$_3$]$^{3-}$ anion. Additionally, Werner also realized that, in the case of non-planar ligands such as ethylenediamine, the property of internal ligand dissymmetry contributes additional sources of chirality that play a role in the decision as to whether a pair of crystallographic entities are truly a racemic pair, or not. Subsequently, Werner and one of his students, Victor L. King [1], demonstrated that solutions of the cations of the above-mentioned iodide rotated the plane of polarized light in water solutions at room temperature. In other words, they were stereochemically robust. Those results were instrumental in Werner receiving the Nobel Prize in 1913. Detailed examples follow.

The packing of the racemic crystals of ($\pm$)-[Co(en)$_3$]OxBr. 3H$_2$O (Ox = oxalato) is shown below in Figure 1. It is a true racemate both in the solid and in solution, which we examine for comparison with the tri-iodide, which is a racemate in solution but a kryptoracemate in crystalline form, as demonstrated below.

Note that the cations in Figure 1: (a) have no chiral atomic centers; and (b) consist of a pair of mirror images exhibiting dissymmetry due to: (1) opposite rotary senses of the ligands about the metal; and (2) the sense of the torsionality of the ethylene ligands (but of utmost importance is the fact that the magnitudes of those ligands' torsional angles are identical and of opposite sign). This defines a crystalline racemate.

By contrast, the same cation, present in solution as a racemate, is found in ($\pm$)-[Co(en)$_3$]I$_3$. H$_2$O crystals as a kryptoracemate and is characterized by the packing presented in Figure 2 below.

A very convenient way to demonstrate this fact is shown in Figure 3 below, which displays an overlay of the ($\pm$)-[Co(en)$_3$]I$_3$. H$_2$O cations as generated in Mercury and implemented in Diamond. Clearly, none of the three rings overlap exactly, as they should in the case of a racemate. In fact, the pair on the left come close to doing so, but the pair of overlapped rings on the right fail miserably. That is the most important feature of a dissymmetric pair of cations in a kryptoracemate of this sort.

How does the above information relate to the case under consideration in this article?

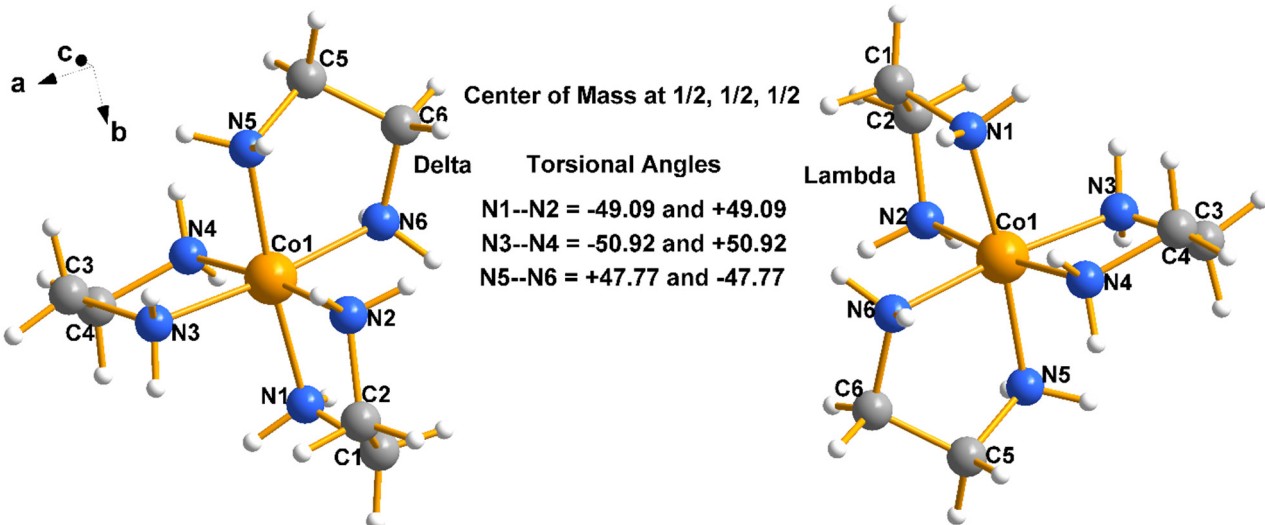

**Figure 1.** In the crystalline form of (±)-[Co(en)$_3$]OxBr. 3H$_2$O, the Co cations are either Lambda (meaning that the sequential sense of rotation about the metal describes an anticlockwise motion (right in the figure)), or a clockwise motion, Delta (left in the figure). The species is a racemate crystallizing in space group C2/c. The torsional angles show that the absolute configurations are exactly opposite in magnitude and sign. Such is not the case in the so-called racemate of the tri-iodide monohydrate, described below.

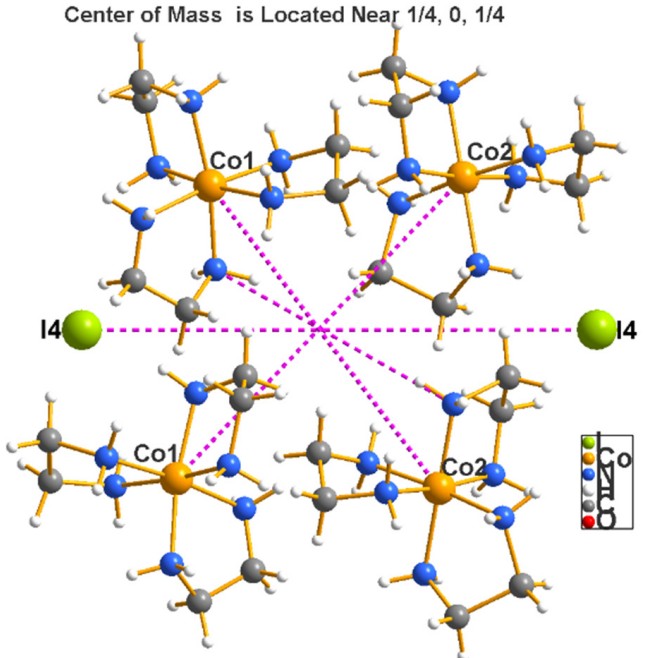

**Figure 2.** The packing of molecules in (±)-[Co(en)$_3$]I$_3$. H$_2$O. Because the center of mass is located near but not at $\frac{1}{4}$, 0, $\frac{1}{4}$, the Co1 and Co2 pairs are no longer mirror images of each other, e.g., Co1 is Lambda and Co2 is Delta. However, the torsional angles are not identical in magnitude and only some differ in sign. Thus, this is no longer a racemate; instead, it is a kryptoracemic pair. Note that only one fragment of the asymmetric unit, the cobalt cation alone, needs to fail with regard to the presence of the required inversion center or mirror plane. This imperfection constitutes the difference between a racemate and a kryptoracemate.

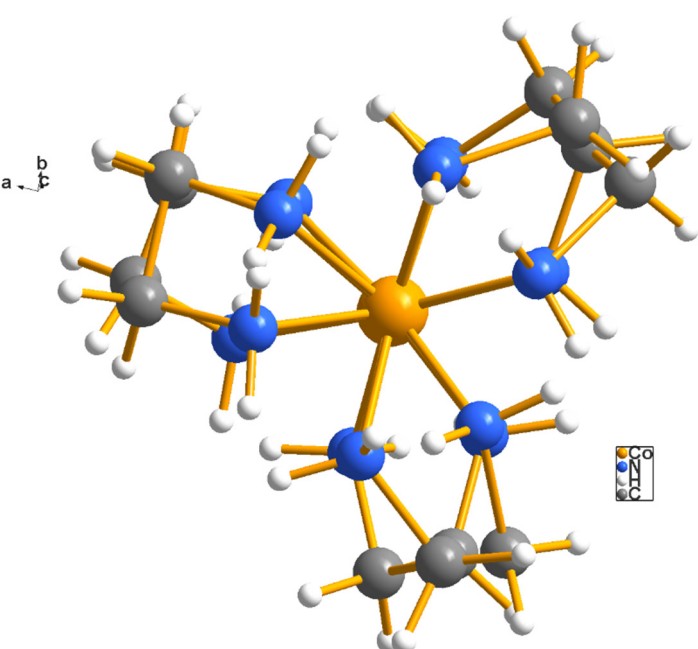

**Figure 3.** Overlay of inverted Co2 onto Co1, optimizing the fit. Drawing generated using Mercury [5] and Diamond [6].

## 2. Results and Discussion

### 2.1. Overview

In Figure 4 below, we show the packing of the structure as described by the coordinates in YETPAP02 [4]. Recall that Z' = 2.0; thus, there are two independent cations and anions which are, for ease of identification, color-identified in this figure.

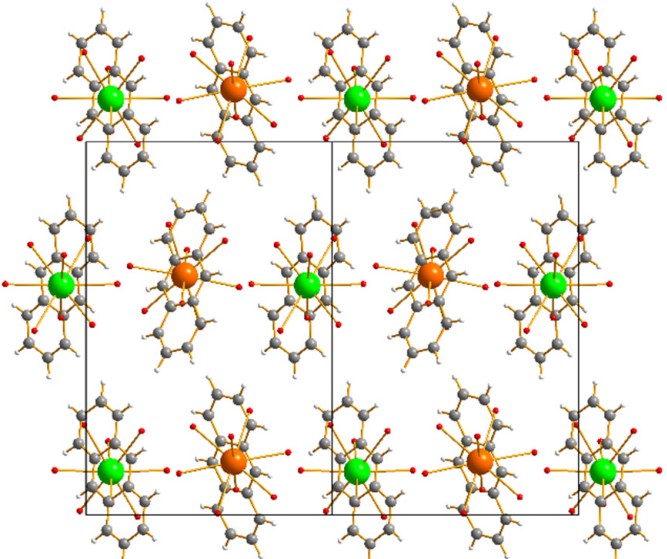

**Figure 4.** Overall view with K1 = green and K2 = orange. If all the atoms were shown, the packing would be so dense as to render the figure nearly useless. By showing (in red) the crown ether oxygen atoms only, one obtains a more informative picture of the packing; the remaining crown ether and furan atoms were eliminated for this view.

By contrast, in Figure 5, we display the packing of the entire contents of the unit cell.

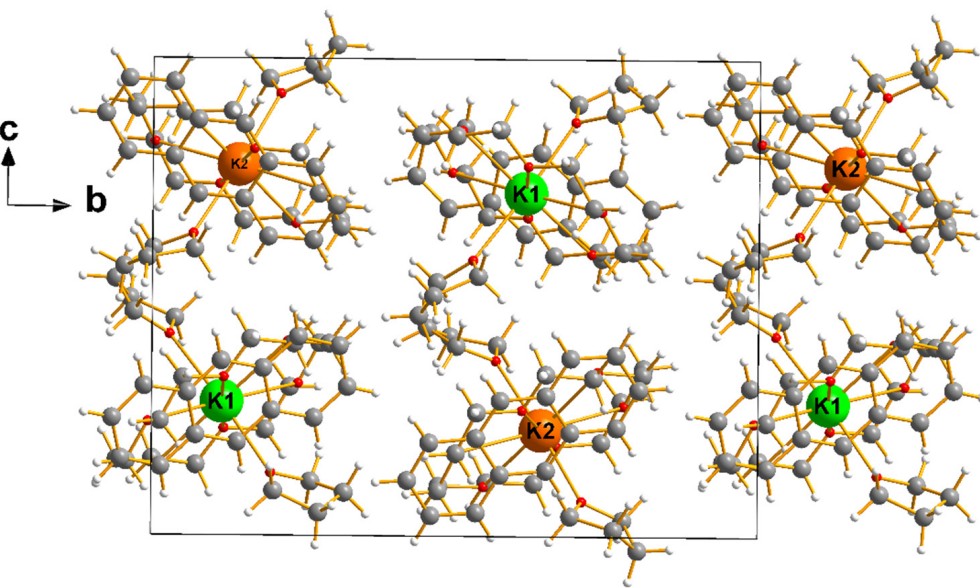

**Figure 5.** This is the best orientation (down the *a*-axis) to show the contents of the unit cell. The oxygen atoms of the axial furan ligands appear as red dots bound to the potassium cations by an orange-colored bond. With the help of Figure 4, one can better understand the relationships between the moieties constituting the crystal's asymmetric unit.

The cation–anion pair constituting the asymmetric unit are nearly identical, as is characteristic of kryptoracemic crystals. Therefore, we show one such pair in Figure 6, in order to illustrate the cation–anion stereochemical relationships present in an asymmetric unit.

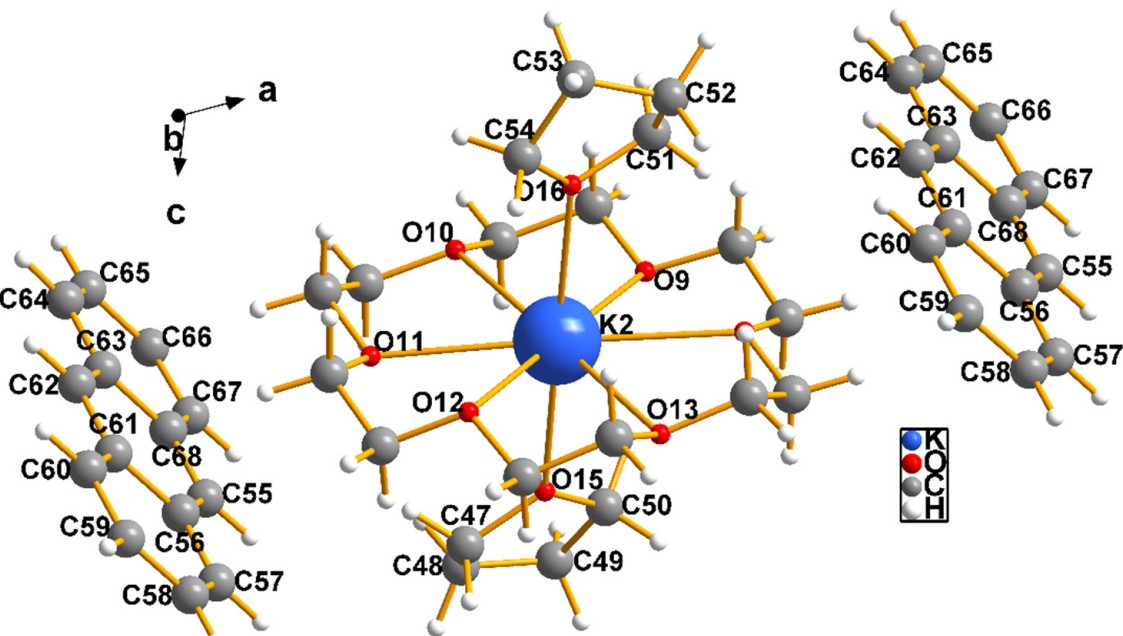

**Figure 6.** Here, the stereochemistry of both the potassium-crown ether cation and the anthracenide anion can be seen without significant overlap. Figure 7 demonstrates that the anions are literally identical, as shown by an overlay picture generated with Mercury [5] and processed in final form with Diamond [6].

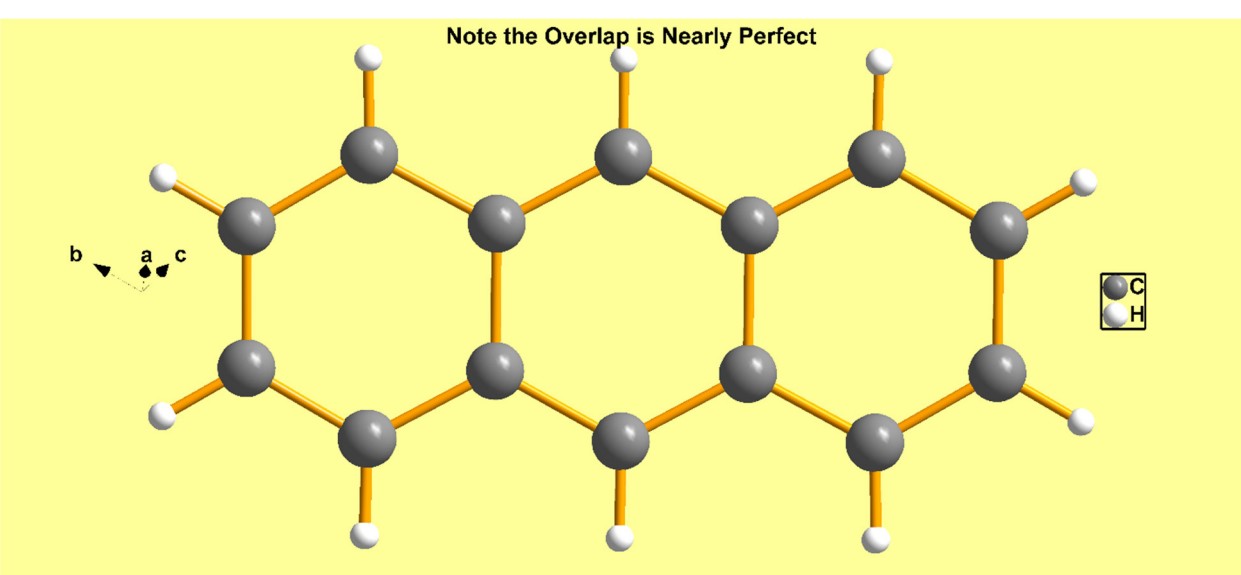

**Figure 7.** At this level of resolution, it is nearly impossible to determine that there are two anions plotted on top of one another. Thus, these fragments do not play any role whatsoever in determining the kryptoracemic nature of these crystals; a role relegated to the cationic species described next in Figure 8.

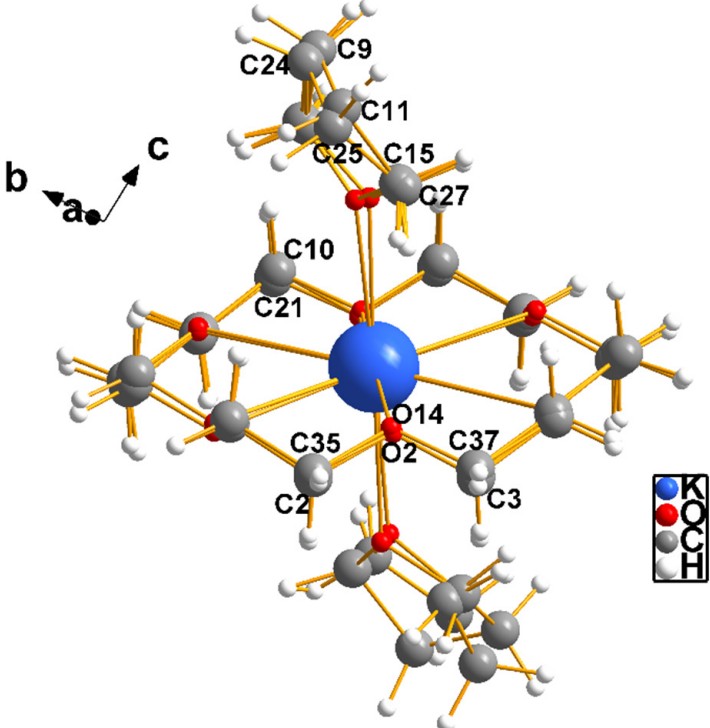

**Figure 8.** As is obvious, the potassium cations and the crown ether ligands fit almost exactly on top of one another; remarkably, even the alkyl hydrogens fit. The main discrepancy in the fit is associated with the axial furan ligands, despite the best efforts of Mercury [5] to fit them using least-squares fitting. This failure is particularly evident in the furan ring at the bottom of the figure, and it is this poor fit and the other more minor misfits that account for the fact that Z' = 2.0, and that the crystal is a kryptoracemate [7–11].

In order to emphasize the fact that the overlay displayed in Figure 8 shows overall discrepancies associated with fitting all the atoms of the cation, the superposition of the two crown ethers alone is shown next in Figure 9.

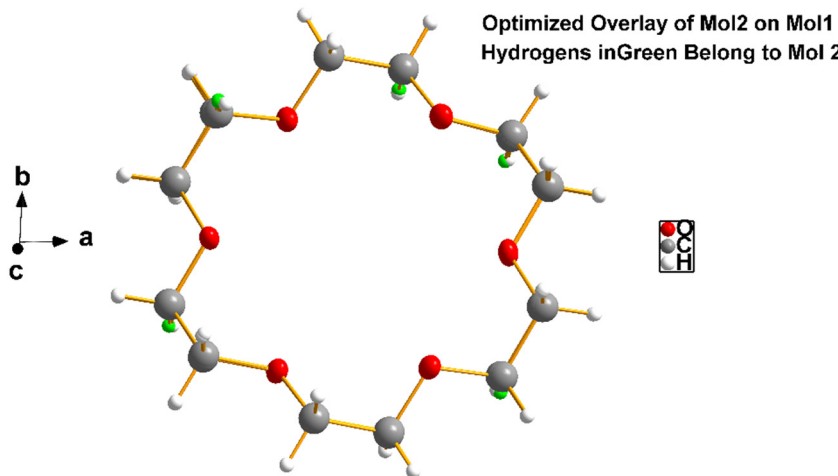

**Figure 9.** It is nearly impossible to tell that there are two ether molecules here superposed on one another, except for the fact that five of the hydrogens differ enough to be discernibly different, as emphasized by the green–white pairs shown above. The idealized geometry of these ether rings is $D_{3d}$.

With the aid of Figure 10, we now begin the discussion of the kryptoracemic nature of this free-radical-containing substance.

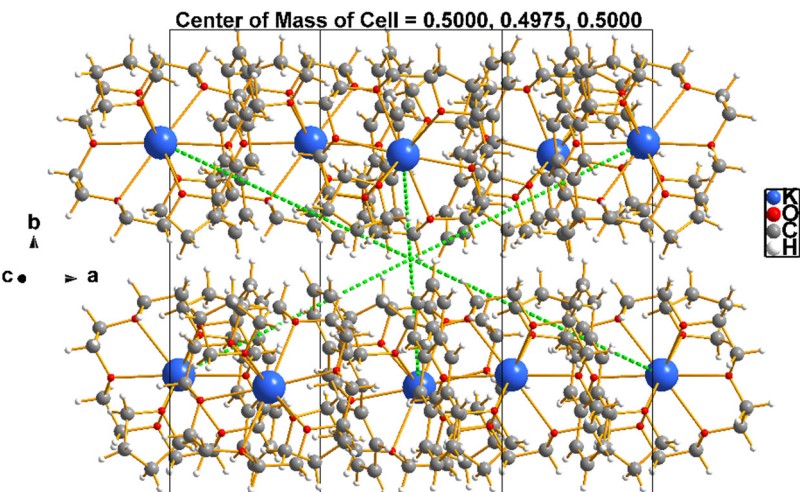

**Figure 10.** The packing of the cations and anions of the unit cell, shown approximately down the *c*-axis. The center of mass for these species is displayed at the intersection of the dotted green lines, located at 0.5000, 0.4975, 0.5000, which is nearly exactly at ½, ½, ½. However, recall that the space group is $P2_1$, where the origin for **y** is arbitrary. This fact is consistent with the kryptoracemic nature of this remarkable substance.

### 2.2. Kryptoracemic Assignment of the Structure: CheckCIF and the Flack x Parameter

CheckCIF indicates that the assignment of $P2_1$ ($Z' = 2.0$) for the space group is correct (see Acknowledgements). Therefore, it is of more than passing interest that the Flack x parameter is listed as 0.42(3), normally indicative of a centrosymmetric crystal. Furthermore, one should recall the remarks of Parsons [12] who stated that: "Flack and Bernardinelli [13] considered how small the standard uncertainty, *u*, of *x* should be before any conclusion

regarding absolute structure can be made. They concluded that even if a compound is known to be enantiopure, the value of *u* should be less than 0.1 before any conclusions regarding absolute structure can be made. If the enantiopurity of the sample is unknown, then the value of *u* should be less than 0.04". In the case of the structural determination by Castillo et al. [4], the relevant value is 0.42(3). It appears, therefore, that the data are precise enough to determine, with very reasonable certainty, that the space group is centrosymmetric, possibly $P2_1/m$. However, if the crystal is a kryptoracemate, the situation is not the usual one encountered in "normal" cases. Recall that: (a) in Figure 9 and related material, it was demonstrated that the entire unit cell is located at a near-perfect inversion center, located within experimental error at ½, ½, ½; (b) at the molecular level, the pair of anthracenide anion radicals are literally identical (see Figure 7 and comments); and (c) Figure 8 and related material demonstrate that the overlay of the two cations shows readily observable but modest deviations from coincidence of oxygens, carbons, and hydrogens; atoms whose anomalous scattering power is extremely small, if important at all, given that the data were collected with Mo radiation. Therefore, it would be surprising if the Flack x parameter showed anything other than a nearly exact centrosymmetric distribution of intensities. The combination of (a) a kryptoracemic crystal with (b) relatively small dissymmetric differences of very low atomic number atoms, and (c) the use of Mo radiation, should produce the observed result.

### 3. Conclusions

(a) The structure of K[(18-crown-6)-bis(tetrahydrofuran)anthracenide], as reported in references [2–4], is that of a radical anion crystallizing as a kryptoracemate [7–11] in space group $P2_1$. Note that the Söhncke space group assignment was made independently, subjected to checkCIF, and published. (b) In addition to the various anthracenides prepared and characterized by Castillo et al. [4], the Cambridge Database [1] reveals that there are five cases of the anthracenide anion radical on record. For the convenience of the readers, a full description of these publications is given as Appendix A below. Not one of these studies reports the anion as crystallizing in a Söhncke space group with Z′ = 2.0.

**Author Contributions:** I.B. and R.A.L. wrote the manuscript together. All authors have read and agreed to the published version of the manuscript.

**Funding:** This research received no external funding.

**Institutional Review Board Statement:** Not applicable.

**Informed Consent Statement:** Not applicable.

**Data Availability Statement:** Not applicable.

**Acknowledgments:** We thank Skye Fortier (University of Texas, El Paso) for useful and important discussions during the preparation of the manuscript, as well as for providing a copy of the results of checkCIF, which he re-ran for us. It agrees that the space group is $P2_1$, as in the initial submission see reference [4], and lists the Flack parameter as 0.42(3), a fact discussed above.

**Conflicts of Interest:** The authors have declared that no competing interest exist.

### Appendix A

BUFSIE

Freeman, P.K.; Hutchinson, L.L. *J. Org. Chem.* **1983**, *48*, 879–881. Magnesium anthracene tetrahydrofuran solvate, $C_{14}H_{10}^{2-}$, $Mg^{2+}$, $3(C_4H_8O)$. Sp. Gr. = no additional information in CSD.

DIHFIJ

Bogdanovic, B.; Janke, N.; Kruger, C.; Mynott, R.; Schlichte, K.; Westeppe, U. *Angew. Chem. Int. Ed.* **1985**, *24*, 960–961. Tris(m₂-Chloro)-hexakis(tetrahydrofuran)-di-magnesium

anthracenide $C_{24}H_{48}Cl_3Mg_2O_6^+$, $C_{14}H_{10}^-$ Sp. Gr. = $C2/c$, Z = 4.0, Z′ = 0.5, R = 3.90, T = 100 K, Diffractometer. Av. Sig. = 0.001–0.005 Å.

MURLIX

Hiley, C.I.; Inglis, K.K.; Zanella, M.; Zhang, J,; Manning, T.D.; Dyer, M.S.; Knaflic, T.; Arcon, D.; Blanc, F.; Prassides, K.; Rosseinsky, M.J. *Inorg. Chem.* **2020**, *59*, 12545–12551. Di-potassium tetracene-diide, $C_{18}H_{12}^{2-}$, 2(K⁺). Sp. Gr. = $P2_1/c$, Z = 4.0, Z′ = 1.0, R = 2.15, T = 298 K, Synchrotron Powder Data. Av. Sig. = 0.001–0.005 Å.

QIBKIY

Ellis, J.E.; Minyaev, M.E.; Nifant'ev, I.E.; Churakov, A.V. *Acta Cryst., Sect. C: Cryst. Struct. Chem.* **2018**, *74*, 769–781. Di-potassium di-scandium tris(anthracene-9,10-di-ide) bis(1,3-diphenyl-cyclopentadienide) tetrahydrofuran solvate, $2(C_{17}H_{13}^-)$, $3(C_{14}H_{10}^{2-})$, $5(C_4H_8O)$, $2(Sc^{3+})$, 2(K⁺). Sp. Gr. = *Ibam*, Z = 4.0, Z′ = 0.25, R = 5.50, T = 150 K, Diffractometer. Av. Sig. = 0.011–0.030 Å. Unfortunately, CSD notes that: "Due to disorder by symmetry, we have represented the structure ionically."

WIPXEY

Bock, H.; Arad, C.; Nather, C.; Havlas, Z. *Chem. Commun.* **1995**, 2393–2394. bis(Diglyme-O,O′,O″)-sodium anthracenide radical, $C_{12}H_{28}NaO_6^+$, $C_{14}H_{10}^-$. Sp. Gr. = $C2/c$, Z = 4.0, Z′ = 0.5, R = 3.62, T = 130 K, Diffractometer. Av. Sig. = 0.001–0.005 Å.

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
