# Peer review of "(18-Crown-6)-bis(tetrahydrofuran)-potassium Anthracenide: The Salt of a Free Radical Anion Crystallizing as a Kryptoracemate"

_chemistry, doi:10.3390/chemistry4010012_

Round 1
Reviewer 1 Report
It can be seen that the author has made adjustments to the article, and a detailed introduction to the background has been presented. The aim of this paper is to prove that K[(18-crown-6)-bis(tetrahydrofuran)anthracenide] is a Kryptoracemate with radical anion crystal. In this, the author is successful, so I think the article can be published.
Author Response
Reviewer (1)
It can be seen that the author has made adjustments to the article, and a detailed introduction to the background has been presented. The aim of this paper is to prove that K[(18-crown-6)-bis(tetrahydrofuran)anthracenide] is a Kryptoracemate with radical anion crystal. In this, the author is successful, so I think the article can be published.
We are happy that reviewer (2) has been able to see that we changed the paper substantially, and that he/she agrees that this is a substantial addition to the literature on this subject.

Reviewer 2 Report
The authors have re-worked the paper again and their presentation is much clearer here. It is much more accessible.
Two comments for the authors to address.
1) If a grad student brought me this crystal structure I agree it was a nice structure, but I would ask them to prove by structure refinement that the structure was did not crystallise in space group P21/m with disorder.
Similarly, I would ask what the intensity statistics suggested about the whether the structure was centrosymmetric.
I appreciate that using someone else's data to do their own work makes it difficult for the authors, but they are in touch with the original author. Original data and refinement should help to support their case. This feels like a very low key way to support their argument as the refinement will not take long.
2) The second point is more philosophical. What is the point in this work? Could the authors make it more accessible for a wider readership by explaining the relevance beyond their niche.
Author Response
Reviewer (2)
The authors have re-worked the paper again and their presentation is much clearer here. It is much more accessible.
Thank you.
Two comments for the authors to address.
1) If a grad student brought me this crystal structure I agree it was a nice structure, but I would ask them to prove by structure refinement that the structure was did not crystallise in space group P21/m with disorder.
The following few pages should answer this question once and for all:
This is a direct quote from the paper by M. Castillo, A. J. Meta-Magaña and Skye Fortier, New J. Chemistry, 40, 1923-26 (2016). Note carefully their argument why there MUST BE two separate, different potassium [18-crown rings]+ in the asymmetric unit:
“Examination of the solid-state structures of 1–12 (see Fig. 1 and ESI†) reveals that nearly all crystallize as non-interacting ion pairs with one notable exception. In 3 (Fig. 1a), the [K(18-c-6)]+ moiety is axially flanked by two bridging [C10H8]-. anions forming a close contact network that gives rise to a 1D coordination polymer. Interestingly, each of the two bridging naphthalenes exhibits a distinct coordination mode. The first naphthalene ligates the potassium cations through η2-binding where the two K–Carene bond distances (avg. 3.13 Å) and K–Carene dihedral angle (120.3o) are indicative of a typical π–cation interaction. The second naphthalene engages each potassium through two longer K–Carene bonds (avg. 3.45 Å) with a notably more obtuse K–Carene dihedral angle (150.3o), parameters that are consistent with agostic interactions between potassium and the C–H bonds of the naphthalene.21 In contrast, it should be noted that [K(18-c-6)-(THF)2][C10H8] exists as a separated ion pair.5 While the exact cause of this structural variation is not known, we attribute the difference to differing crystallization methods and conditions (ESI†).5
---------------------------------------------------------------------------------
Separately, here is an analysis of the geometry of the torsional angles of the crown ether rings to additionally justify the fact that there are two separate ones (showing the kryptoracemate in our paper); this necessarily negates the space group as possibly being P21/m with disordered THF molecules.
Below is a Figure showing the two rings referred to above:
Labeling of the atoms in the Castillo paper: Torsional Angles:
K1 Ring K2 Ring
O1C4C3O2 -62.58 O9C22C21O11 -60.44
O2C2C6O4 -70.14 O11C26C31O10 +64.46
O4C16C7O6 -70.57 O10C28C34O12 -66.75
O6C1C10O5 +66.70 O12C36C37O14 +61.30
O5C8C5O3 +66.23 O14C35C29O13 -66.62
O3C12C14O1 +67.03 O13C32C23O9 +70.04
As is evident, the torsional angles are not those of a PERFECT racemic pair. Therefore, MERCURY tries, as best it can to match them by INVERTING them and by a least-squares fit. The result is shown in Fig. 9 of the manuscript. Note that such operation results in an attractive fit! But, it is still imperfect; thus, an attempt to claim that the space group is P21/m is totally unacceptable.
IN ADDITION:
Figure 8 of our manuscript shows that the fit is even worse when the axial furans are added to complete the entire K[(18-crown-6)-bis(tetrahydrofuran)] cation. That VERY imperfect fit by the axial furan ligands dooms any attempt to assign the space group as P21/m to the data sets by both Kochi (the original data) and Castillo (the current one), which, by the way, were examined by Check.Cif, referees, editors, and published separately in different journals and entered into CSD without any evidence of challenging the space group assignment as P21.
To re-iterate, and to make it absolutely clear:
The compound in question was synthesized independently by three groups; and the structure was determined independently by those same groups using different crystals, different diffractometers, and published after being dully refereed. Clearly, they were acceptable inasmuch as they were published and entered into CSD, implying that they were examined by Check.Cif, and the R factors were uniformly attractive, especially for TWO of them, as follows:
YETPAP: S. V. Rosokha, J. K. Kochi; R= 4.91, at 123 K
YETPAP02: M. Castillo, A. J. Metta-Magana, S. Fortier, R = 4.44 at 100 K.
Could the reviewer (a) tell us why any of this has to be repeated by us? (b) has this outrageous request ever been made from anyone he knows of (c) what would the reviewer think if a reviewer were to make the same requests in a contribution of his? Similarly, what would the intensity statistics suggest about whether the structure was centrosymmetric?
On p. 9 of our revised Ms, Section 2.2, we describe exactly why this CANNOT be centrosymmetric, and MUST be a kryptoracemate.
I appreciate that using someone else's data to do their own work makes it difficult for the authors, but they are in touch with the original author. Original data and refinement should help to support their case. This feels like a very low key way to support their argument as the refinement will not take long.
Of what use is repeating the refinement? The data have been accepted and gone through rigorous testing through Check.Cif and CCDC.
2) The second point is more philosophical. What is the point in this work? Could the authors make it more accessible for a wider readership by explaining the relevance beyond their niche.
This paper is for professionals who read PROFESSIONAL JOURNALS. Maybe, one of these days, we will write a diluted version for the Journal of Chemical Education.